# Mimicking the Liver Sinusoidal Endothelial Cell Niche In Vitro to Enhance Fenestration in a Genetic Model of Systemic Inflammation

**DOI:** 10.3390/cells14080621

**Published:** 2025-04-21

**Authors:** Dibakar Borah, Oliwia Blacharczyk, Karolina Szafranska, Izabela Czyzynska-Cichon, Sara Metwally, Konrad Szymanowski, Wolfgang Hübner, Jerzy Kotlinowski, Ewelina Dobosz, Peter McCourt, Thomas Huser, Malgorzata Lekka, Bartlomiej Zapotoczny

**Affiliations:** 1Department of Biophysical Microstructures, Institute of Nuclear Physics, Polish Academy of Sciences, ul. Radzikowskiego 142, 31-342 Krakow, Poland; dborah@ifj.edu.pl (D.B.); sara.metwally@ifj.edu.pl (S.M.); malgorzata.lekka@ifj.edu.pl (M.L.); 2Vascular Biology Research Group, Department of Medical Biology University of Tromsø—The Arctic University of Norway, 9019 Tromsø, Norway; karolina.szafranska@uit.no (K.S.); peter.mccourt@uit.no (P.M.); 3Jagiellonian Centre for Experimental Therapeutics (JCET), Jagiellonian University, Bobrzynskiego 14, 30-348 Krakow, Poland; iza.czyzynska@jcet.eu; 4Biomolecular Photonics, Faculty of Physics, Bielefeld University, 33615 Bielefeld, Germany; whubner@mac.com (W.H.); thomas.huser@physik.uni-bielefeld.de (T.H.); 5Department of General Biochemistry, Faculty of Biochemistry, Biophysics and Biotechnology, Jagiellonian University, Gronostajowa 7, 30-387 Krakow, Poland; j.kotlinowski@uj.edu.pl; 6Department of Microbiology, Faculty of Biochemistry, Biophysics and Biotechnology, Jagiellonian University, Gronostajowa 7, 30-387 Krakow, Poland; e.dobosz@uj.edu.pl

**Keywords:** liver sinusoidal endothelial cells, polyacrylamide, atomic force microscopy, elastic properties, fenestrations, actin cytoskeleton, mechanotransduction

## Abstract

Liver sinusoidal endothelial cells (LSECs) play a crucial role in hepatic homeostasis, clearance, and microcirculatory regulation. Their fenestrations—patent transcellular pores—are essential for proper liver function, yet disappear in pathological conditions such as liver fibrosis and inflammation through a process known as defenestration. Defenestrated sinusoids are often linked to the liver stiffening that occurs through mechanotransduction-regulated processes. We performed a detailed characterization of polyacrylamide (PAA) hydrogels using atomic force microscopy (AFM), rheometry, scanning electron microscopy, and fluorescence microscopy to assess their potential as biomimetic substrates for LSECs. We additionally implemented AFM; quantitative fluorescence microscopy, including high-resolution structured illumination microscopy (HR-SIM); and an endocytosis assay to characterize the morphology and function of LSECs. Our results revealed significant local variations in hydrogel stiffness and differences in pore sizes. The primary LSECs cultured on these substrates had a range of stiffnesses and were analyzed with regard to their number of fenestrations, cytoskeletal organization, and endocytic function. To explore mechanotransduction in inflammatory liver disease, we investigated LSECs from a genetic model of systemic inflammation triggered by the deletion of Mcpip1 in myeloid leukocytes and examined their ability to restore their fenestrations on soft substrates. Our study demonstrates the beneficial effect of soft hydrogels on LSECs. Control cells exhibited a similar fenestrated morphology and function compared to cells cultured on plastic substrates. However, the pathological LSECs from the genetic model of systemic inflammation regained their fenestrations when cultured on soft hydrogels. This observation supports previous findings on the beneficial effects of soft substrates on LSEC fenestration status.

## 1. Introduction

Mechanotransduction in the liver plays a crucial role in regulating hepatic function, as liver cells constantly sense and respond to mechanical signaling from their microenvironment [1]. These signals result from the blood flow, extracellular matrix (ECM) stiffness, and cellular interactions, influencing processes such as metabolism, regeneration, and fibrosis and altering the liver’s architecture [2]. The liver’s unique microvascular network of sinusoids facilitates the efficient reciprocal exchange of metabolic substrates between circulating blood and hepatocytes, playing a crucial role in homeostasis and detoxification [3,4]. Liver sinusoidal endothelial cells (LSECs) line the hepatic sinusoids, creating a perforated physical barrier between the blood and the liver parenchyma [5]. LSECs have evolved unique properties such as the lack of a basal lamina and the presence of patent transcellular pores, called fenestrations, allowing them to participate in the liver clearance system [6]. Fenestrations with diameters in the range of 50–350 nm ensure the passive, size-regulated bidirectional transport of substances between the liver parenchyma and bloodstream. In addition, fenestrations facilitate direct interactions between circulating immune cells and underlying hepatocytes [7]. Unlike in most tissues, where immune cells must extravasate to reach their targets, liver-resident effector CD8+ T cells can probe hepatocytes directly from the intravascular space by extending cytoplasmic protrusions through the fenestrations of LSECs [8,9]. However, during liver fibrosis, sinusoidal defenestration and capillarization [10] impair this process, reducing immune surveillance and facilitating disease progression. Finally, LSECs express a number of scavenger receptors which, together with the highest receptor-mediated endocytic capacity in the body, constitute the scavenging system facilitating the efficient clearance of bloodborne waste macromolecules [11,12]. Both the porosity and scavenging properties of LSECs decrease with aging and with the development of liver diseases [3,6,13]. Losing fenestrations is accompanied by alterations in the cardiovascular system and the efficiency of drug clearance and the metabolism and hyperlipidemia [14,15]. Thus, multiple studies have addressed restoring fenestrations pharmacologically in defenestrated LSECs to improve liver function [6,16].

Macroscopically, it has been demonstrated that the liver stiffens with the level of pathology and that a reduction in liver stiffness can be a good prognostic marker [17]. The stiffening of the liver is mainly observed due to the deposition of scar tissue; however, it was demonstrated that altered biomechanics are preserved in cells isolated from diseased livers [18,19]. Studies of liver cells suggest that substrate stiffness has a profound influence on the physiology of endothelial cells [20], including that of LSECs [20,21,22,23]. These authors demonstrated that the number of fenestrations increases when these cells are cultured on soft substrates and, in particular, LSECs from the model of liver cirrhosis restored their functionality when placed in a soft environment. These findings highlight the critical interplay between ECM stiffness, mechanotransduction pathways, and endothelial function in liver health and disease. As mechanosensitive cells [24], LSECs respond to substrates’ mechanical properties through cytoskeletal remodeling and signaling, which dictate their structural integrity and clearance efficiency. By integrating knowledge from material science and cellular mechanobiology, efforts are underway to create microenvironments that preserve or restore LSEC fenestrations, thereby improving overall liver function and offering new therapeutic strategies for hepatic disorders.

In this study, we synthesized, functionalized, and characterized polyacrylamide (PAA) hydrogels that mimic the LSEC niche. We characterized the morphology and biomechanical properties of the LSECs on selected hydrogels. Finally, we examined the therapeutic effect of a soft environment on LSEC fenestrations in the model of systemic inflammation triggered by the deletion of Mcpip1 in myeloid leukocytes (Mcpip1^fl/fl^LysM^Cre^) [25]. Our recent study showed that LSECs isolated from 6-month-old Mcpip1^fl/fl^LysM^Cre^ mice exhibited severe defenestration, which was only partially reversible using the actin-depolymerizing toxin cytochalasin B [18]. In the current study, we demonstrate a similar, positive effect on LSECs in this genetic model when they are cultured on soft substrates.

## 2. Materials and Methods

### 2.1. Preparation of Substrates

#### 2.1.1. Preparation of Hydrophilic Surfaces

Round slides with diameters of 13 mm or 25 mm, as well as 8-well plates, were used as substrates for hydrogel synthesis. The slides were cleaned in 1 M NaOH (Avantor, Gdansk, Poland) for 30 min before use. They were then washed in deionized water 4 times for 5 min each and air-dried. Cleaned slides or well plates were transferred to a desiccator and functionalized with vapors of 3-aminopropyltriethoxysilane (APTES; Merck, Poznan, Poland) under vacuum for 2 h. Then, the slides were additionally functionalized with 0.5% glutaraldehyde (Merck, Poznan, Poland) for 1 h. The slides were washed 4 times in deionized water for 5 min each and air-dried.

#### 2.1.2. Preparation of Hydrophobic Slides

Coverslips (24 mm × 24 mm) were dipped in 5% SurfaSil (Thermo Scientific, Waltham, MA, USA) solution for 10 s and washed with acetone (Avantor Performance Materials Poland S.A., Gdansk, Poland) and 2× methanol (Avantor, Gdansk, Poland) and then air-dried.

#### 2.1.3. Synthesis of Polyacrylamide Hydrogels (PAAs)

The hydrogel preparation method was adapted from a previous report [26]. Mixtures containing 40% acrylamide (Merck, Poznan, Poland), 2% bisacrylamide (Fisher BioReagents, POL-AURA, Gdansk, Poland), and deionized water were prepared in the proportions described in Table 1. Freshly prepared 10% ammonium persulfate (APS; Sigma-Aldrich, Darmstadt, Germany) and N,N,N′,N′-tetramethylethane-1,2-diamine (TEMED; Sigma-Aldrich, Darmstadt, Germany) were then added to induce polymerization. The 20 µL solution was quickly transferred to the hydrophobic slides and covered with a hydrophilic slide. After the thickening of the hydrogels, the hydrophilic slides were separated from the hydrophobic slides and washed in phosphate-buffered saline (PBS; Sigma-Aldrich, Darmstadt, Germany). In parallel, 3.2 mL of each solution was placed in a single well of the 8-well plate to create a ~2 mm thick hydrogel for rheometer measurements. The prepared hydrogels were stored at 4 °C in PBS with antibiotics for hydration and to prevent bacterial growth. The hydrogels were used within 2 weeks (for AFM and optical microscopy) or 2 days (for the rheometer).

#### 2.1.4. Coating of Functionalized Hydrogels with Proteins

The hydrogels were activated using 3,4-dihydroxy-L-phenylalanine (L-DOPA; Sigma-Aldrich, St. Louis, MO, USA). First, a 2 mg/mL solution of L-DOPA was prepared in 10 mM Tris-buffered saline (Sigma-Aldrich, St. Louis, MO, USA) at pH 10 for 30 min in the dark and sterilized by passing it through a 0.22 µm cellulose acetate filter, as described elsewhere [27]. The PAA hydrogels were immersed in L-DOPA solution and left in the dark for 30 min. The hydrogels were then washed twice with PBS to remove residual unbound L-DOPA. ECM protein Type III Collagen Solution (Advanced BioMatrix, San Diego, CA, USA) was applied immediately after the hydrogels were functionalized with L-DOPA. Briefly, collagen III (0.1 mg/mL) was applied to the surface of the hydrogels and incubated for 2 h at room temperature. After this time, the samples were washed 3 times with PBS to remove unbound protein. They were stored at 4 °C for up to 3 days before use or up to 1 day for the rheometry tests. In parallel, prepared hydrogels were compared with commercially available PAA hydrogels functionalized with collagen type I (Cell Guidance Systems), which had an elasticity ranging from 0.1 to 100 kPa.

### 2.2. Isolation and Culture of LSECs

LSECs were isolated either from wild-type male C57BL/6 mice or, for the final experiments, from 6-month-old mice with a deletion of the Zc3h12a gene encoding Mcpip1 (monocyte chemoattractant protein-induced protein 1) in myeloid leukocytes (Mcpip1^fl/fl^ LysM^Cre^) and from control Mcpip1^fl/fl^ mice. A description of the genotyping is described elsewhere [25]. The procedure for isolating LSECs has been described in detail elsewhere [28,29]. Briefly, male mice were anesthetized using a mixture of ketamine/xylazine before liver perfusion and digestion were performed using LiberaseTM (Roche, Basel, Switzerland). Parenchymal cells were removed by differential centrifugation. LSECs were separated using immunomagnetic beads conjugated with anti CD146 antibodies (MACS, Miltenyi Biotec, Bergisch Gladbach, Germany). Isolated LSECs were seeded on the prepared substrates (200,000 cells per sample). Cell cultures were gently washed 15 min after seeding to remove any non-adherent cells. LSECs were cultured for >6 h in EGM-2 medium (Lonza, Basel, Switzerland) before starting AFM measurements or overnight (18–20 h) before fixation for other tests.

### 2.3. Atomic Force Spectroscopy and Microscopy

Atomic force microscopy (AFM) was used to measure the topography and nanomechanical properties of the PAA hydrogels and LSECs. XE-120 (Park Systems, Suwon, Republic of Korea) and Nanowizard IV (Bruker-JPK Instruments, Berlin, Germany) were used in this study. The detailed procedures for calculating the apparent Young’s modulus [30,31], LSEC porosity [32,33], and the deformability of fenestrations [18] have been presented in earlier works and are briefly summarized below.

#### 2.3.1. Imaging of Hydrogel and LSEC Topography

For assessing LSEC porosity, imaging was generated using SCM-PIC-V2 (Bruker) cantilevers with a nominal spring constant k = 0.1 N/m and nominal tip apex radius of 25 nm in their Quantitative Imaging (QI) mode. In QI mode, an image is acquired by generating multiple force curves, creating a dense map which results in each pixel (px) of the image being described by an individual force–distance curve. The image can be reconstructed for the selected loading force, resulting in complex information about the sample topography that corresponds to the selected loading force. Multiple images can be reconstructed from one scan, each representing the topography of the selected loading force. The load force was adjusted for the individual cantilevers to achieve the best spatial resolution without the distortion of fenestrations (or hydrogels) and was in the range of 200–350 pN. The length of the force curves (the *z* range) and the acquisition speed were 950–1050 nm (up to 3000 nm for soft hydrogels) and 120–140 µm/s, respectively. Image acquisition lasted 35–65 min, depending on the scan’s frame size and pixel density. As established before, we measured large areas covering at least one LSEC, which were not smaller than 1000 µm^2^ [33]. Measurements were conducted at 25 °C for fixed LSECs and hydrogels and at 37 °C for living LSECs, using a PetriDish Heater™ (Bruker-JPK Instruments, Berlin, Germany).

#### 2.3.2. Deformability of LSEC Fenestrations

To assess the deformability of the fenestrations, QI images of LSEC sieve plates were acquired using a method similar to that described for LSEC topography. Each biological replicate used a new cantilever to prevent alterations due to changes in tip geometry. A constant load force of 350 pN was applied during imaging. Images were then reconstructed for load forces of 170 pN and 300 pN, which correspond to ~50% and ~80% of the maximum load force, respectively. The 170 pN force was chosen as the minimal value that reduced noise while allowing a clear visualization of the fenestrations. The fenestration diameters in both images were manually measured in the fast scan axis (time per line 1.0–2.5 s) using Fiji [34]. The enlargement of individual fenestrations was calculated as the ratio of the diameter measured at 300 pN to that measured at 170 pN and expressed as a fold change. The pixel size used in the measurements varied from 20 to 27 nm (typical image size: 3.5 µm × 3.5 µm) and the image resolution from 128 × 128 px to 175 × 175 px, which indicates error in the measurements.

#### 2.3.3. Force Spectroscopy

The AFM force spectroscopy mode was used to determine the apparent elastic modulus (apparent Young’s modulus) of the investigated objects. Force–distance curves were acquired from the central area of the cell (or in randomly selected areas of the hydrogel) with sharp or hemispherical probes in force–volume mode. Sharp silicon nitride probes (MLCT-BIO-DC, Bruker) were used with a load force that resulted in an indentation of 1000–1500 nm to investigate the apparent Young’s modulus. Additionally, parts of the experiments were repeated using pre-calibrated, hemispherical silicon nitride probes, MLCT-SPH-5UM (Bruker), with a tip radius of 5.0 µm and tip height of 25 µm. Measurements of the hydrogels were performed with a load force that resulted in an indentation of 900–1000 nm. The acquisition speed was always set to 8.0 µm/s. The elastic modulus was calculated using JPK Processing Software (V6.4.21) according to the Hertz–Sneddon model of contact mechanics. Before measurements, a calibration force–distance curve was acquired on a glass (non-deformable) surface for the non-calibrated cantilevers. The spring constant was calibrated using a thermal tune [35].

### 2.4. Fluorescence Microscopy

#### 2.4.1. Protein Staining of the Substrates Studied

Protein-coated hydrogels were immunolabeled with mouse anti-collagen monoclonal antibody (1:500; MAB3392, Sigma-Aldrich) overnight. Then, goat anti-mouse antibody conjugated with AlexaFluor555 (1:1000 in PBS; Invitrogen, Thermo Fisher Scientific, Waltham, MA, USA) was applied for 2 h and the hydrogels were then rinsed with PBS.

#### 2.4.2. Staining of LSECs

LSECs were fixed with 3.7% paraformaldehyde (30 min) and permeabilized using 0.2% Triton-X100 for 5 min. Cells were labeled with monoclonal anti-α-tubulin antibodies (Sigma-Aldrich) and stained with anti-mouse antibodies conjugated with Alexa Fluor 555 (goat anti-mouse IgG (H + L) cross-adsorbed secondary antibody, Invitrogen, Thermo Fisher Scientific, Waltham, MA, USA). Next, the samples were labeled with phalloidin-AlexaFluor488 (1:200; Invitrogen, Thermo Fisher Scientific, Waltham, MA, USA) in PBS for 45 min and Hoechst 34580 (1:5000; Sigma-Aldrich, Poznan, Poland) for 10 min at room temperature. Samples were washed with PBS for 5 min 3–7 times. Finally, samples were mounted on standard microscopy slides using a Prolong Diamond mounting medium.

#### 2.4.3. Fluorescence Microscopy and Structured Illumination Microscopy

Images were acquired using a 40× objective (NA = 0.6) or 100× objective (NA = 1.2) used with an inverted microscope (IX83, Olympus, Tokyo, Japan), with a mercury lamp as a light source and a Prime BSI Express Scientific CMOS camera (01-prime-BSI-EXP).

Super-resolved fluorescence microscopy images were acquired using a 60× objective lens (NA 1.49) on a 3D structured illumination fluorescence microscope (DeltaVision|OMX v4, Cytiva, Marlborough, MA, USA). Images of AlexaFluor 488-stained actin were acquired after excitation with a diode-pumped solid-state laser at 488 nm, while Alexa Fluor 555 images of tubulin required excitation at 532 nm. The emitted-fluorescence images were collected after passing through narrow bandpass filters centered on the main emission wavelength by individual sCMOS cameras. Figures were initially designed in the OMERO.figure (V4.4.3) web module [36], where all image intensities were adjusted linearly for optimal representation.

### 2.5. Scanning Electron Microscopy (SEM)

PAA hydrogel solution (1 mL) was transferred into 24-well plates and incubated with dH_2_O for 48 h at 4 °C after polymerization. Next, samples were fixed with 0.5% glutaraldehyde solution (Avantor Performance Materials, Gliwice, Poland) for 60 min, followed by rinsing in dH_2_O. Individual hydrogel samples were cut from the plates using a biopsy punch with an 8 mm diameter and transferred into new well plates. Prior to SEM imaging, samples were dehydrated by freezing them in liquid nitrogen and lyophilizing them in a Christ alpha 1–2 freeze-dryer (Martin Christ GmbH, Osterode am Harz, Germany) at −80 °C under a controlled pressure of ~10^−5^ Pa for 12 h. The dehydrated samples were coated with a 5 nm gold layer using rotary-pumped sputter coating (Q150RS, Quorum Technologies, Laughton, UK). Microstructure imaging was performed using a scanning electron microscope (Merlin Gemini II, (Carl Zeiss AG, Oberkochen, Germany) working at a current of 130 pA and voltage of 7 kV.

### 2.6. Rheology of Hydrogels

The PAA hydrogels were polymerized in 8-well plates and collagen-coated as described above. The thickness of the sample was kept constant, at 1.5 mm. Individual samples were cut from plates with a biopsy punch (of an 8 mm diameter) corresponding to the diameter of the upper plate of the rheometer. Rheological measurements in oscillation mode were performed at 37 °C using a parallel plate rotational rheometer MRC302 (Anton Paar), for which a liquid container was filled with dH_2_O to avoid sample dehydration during measurement. To determine the hydrogel’s linear viscoelastic range (LVER), amplitude sweep measurements were performed, applying a shear strain *γ* = 0.01 ÷ 100% and constant angular frequency of *ω* = 1 rad/s. Three independent measurements were performed for each sample, and the results are presented as an average value with an error based on standard deviation. LVER describes the part of the storage *G*′ and loss *G*″ moduli ranges in which the applied stresses are insufficient to induce the structural breakdown (yielding) of the hydrogel and hence its microstructural properties are captured. The LVER was determined by fitting a linear function to the parallel region of the shear strain curves before the decrease in the storage modulus (10% deviation from its plateau value). Based on LVER, the average *G*′ value was calculated to estimate the Young’s modulus, in a manner similar to that described elsewhere [37]. We consider the hydrogels to be perfectly incompressible and linearly elastic, showing negligible viscous behavior. This assumption results in the shear modulus *G* being interchangeable with *G*′, while the relationship between G and elastic modulus E for a linear elastic material can be described as follows:$$G=\frac{E}{2(1+\nu )}$$

When assuming Poisson’s ratio means ν=0.5 (similar to the assumption in AFM),$$E=2G\;\left(1+\nu \right)=3\;G=3\;G^{\prime }$$

### 2.7. Endocytosis Assay

The LSECs were seeded in 24-well plates containing PAA hydrogels (2/8/100 kPa; Cell Guidance Systems, Cambridge, UK) or plastic-coated with collagen type I (PureCol), with 500,000 cells seeded in each well. For the endocytosis experiments [11,38], the cell media was supplemented with 1% human serum albumin, 30 ng/mL of radiolabeled formaldehyde-treated serum albumin (^125^I-FSA), and various concentrations of unlabeled FSA (0–240 µg/mL) and incubated for 2 h. Then, the cell media were removed and undigested ligands were precipitated using 20% trichloroacetic acid to calculate the degradation level. To calculate the number of cells in each well, the cell nuclei were stained (1:1000 Hoechst 33258 for 20 min at 37 °C). Images obtained using fluorescence microscopy were analyzed using simple threshold-based segmentation in *Fiji* [34]. Cells were subsequently lysed with 1% sodium dodecyl sulfate (SDS) to analyze the cell-associated fraction of FSA. All experiments were performed with 3 bioreplicates/animals and 2 technical replicates per experiment. Total endocytosis (the sum of the cell-associated and degraded fractions) was normalized to the calculated number of cells in each well.

### 2.8. Statistical Analysis

The statistical significance of the results was calculated using Student’s *t*-test for unpaired data. The box charts show the mean (line) and the box range represents the standard deviation; the whiskers indicate the 5 and 95% data ranges. A *p*-value less than 0.05 was considered significant. * *p* < 0.05; ** *p* < 0.01; *** *p* < 0.001.

## 3. Results

The results have been organized into three sections. Firstly, we characterized two types of synthesized hydrogels—soft (0.8 ± 0.4 kPa) and stiff (52 ± 8 kPa)—revealing their surface topography, mechanical properties, and protein coverage. We used plastic/glass substrates (>1 GPa) as a reference. Soft hydrogels were prepared to mimic a healthy liver niche in the cell culture and stiff hydrogels to be PAA controls with high stiffness, limiting mechanotransduction responses; glass/plastic substrates were used as reference standard culture substrates. Then, we seeded LSECs onto these hydrogels and characterized their cytoskeleton, biomechanics, surface area, cell nucleus size, and endocytosis efficiency. In addition, a comparison of their wide elasticity range to that of the commercial PAA hydrogels was carried out. Finally, we used the soft and stiff hydrogels to culture LSECs isolated from Mcpip1^fl/fl^ LysM^Cre^ mice to assess the effect of mechanotransduction on the number of fenestrations in the LSECs within a genetic model of systemic inflammation.

### 3.1. PAA Hydrogels Functionalized with Collagen III

Firstly, the synthesized hydrogels were dehydrated and characterized using SEM (Figure 1A). Despite the partial collapse of the hydrogels’ structure and the coalescence of some of the pores after dehydration, the characteristic PAA morphology of both hydrogels could be depicted. As a result, the differences between the hydrogel types could be clearly observed. Soft hydrogels were characterized by a hierarchical structure in their scaffolds, with both large and fine voids present. In contrast, the stiff hydrogels had a more homogeneous structure; large voids were still observed, but their microstructure was less porous. Additionally, we used AFM to study the topography and nanomechanics of hydrated hydrogel samples. The topography of both hydrogels was flat, and their peak-to-trough amplitude was lower than 50 nm. Similarly to the SEM, for the stiff hydrogels we observed heterogeneity in the occurrence of pores on the surface of the gels, but, in general, a network of pores with diameters from tens of nanometers up to a few micrometers (0.7 ± 0.3 µm) was observed (Figure 1B). Larger streaks, reflecting the glass coverslip’s topography, imprinted in the gel were also observed. Additional imaging was performed on the soft hydrogels (Appendix A). The AFM imaging was strongly hampered by the low stiffness of these hydrogels. By using the Hertz–Sneddon formula [39] with the applied imaging parameters of a loading force, F = 100 pN, assuming the Young’s modulus of the hydrogel E = 0.8 kPa, Poisson’s ratio ν = 0.5, and a tip radius of 20 nm, the resulting indentation, δ, became approx. 0.6 µm. This means that the acquired AFM image corresponds to a highly deformed surface, hampering the visualization of the detailed structure of soft hydrogels to a nanometer resolution (Appendix A). Still, our results from the SEM and AFM clearly demonstrate that the topography of the synthesized hydrogels is flat and suitable for cell culture. Moreover, any local differences in the sample topography are negligible compared to the cell size. After preparing the flat hydrogels, our aim was to produce a thin layer of collagen on the hydrogels to provide them with good adhesive properties, allowing the LSECs to spread on the surface, while limiting its thickness to prevent the creation of a separate layer with its own mechanical properties. Using 0.1 mg/mL of collagen III (see Section 2 “Materials and Methods”), we obtained the desired thin layer of collagen, such that individual collagen fibers were present on the surface of the hydrogel (Figure 1B).

Studies of their mechanical properties using AFM (Figure 1C) and the rheometer (Figure 1D) allowed us to assess the apparent Young’s modulus of the hydrogels. In AFM, we tested hemispherical cantilevers and compared the data with thse from pyramidal (traditional) cantilevers. Soft and stiff hydrogels had an apparent Young’s modulus of 0.8 ± 0.4 kPa and 52 ± 8 kPa, respectively. Our results indicated that hemispherical cantilevers provided a lower scattering of the results (Appendix A). Nevertheless, the lower pressure exerted by the large tip resulted in a shallow indentation of up to 300 nm. To obtain a consistent indentation of 1 µm, allowing for a more accurate comparison of the results, we used stiff cantilevers with pyramidal tips (MLCT-BIO-DC, type F) for the stiff hydrogels and soft cantilevers with hemispherical tips (MLCT-SPH-5UM, type D) for the soft hydrogels; all cantilevers had the same V-shaped geometry. We observed a slight decrease in the apparent Young’s modulus values of the stiff hydrogels when collagen was deposited on their surfaces (Figure 1C). Considering collagen hydrogels, in low concentrations, tend to be soft (usually below 100 Pa [40]) this reduction in hydrogel stiffness is expected to be profound for the stiff hydrogels and non-significant for the soft hydrogels, which is in line with our results.

The linear viscoelastic region (LVER) defined in the shear strain curves obtained from the rheological data allowed us to calculate Young’s modulus. The values obtained using the rheometer were usually lower than those obtained for AFM and equal 0.39 ± 0.15 kPa and 26.3 ± 10 kPa for the soft and stiff hydrogels, respectively (n = 3). Representative results are presented in Figure 1D. The discrepancies between the techniques originate from the different scales of imaging used, localized measurements at the hydrogel surface in AFM, and the accuracy of the methods. We observed a similar trend in the stiffening of soft hydrogels after their functionalization with collagen, which was more pronounced in the rheometer data than in those from AFM. We did not expect to observe such a large change in macroscopical measurements with the rheometer. To validate this observation, we performed additional measurements of samples activated with L-DOPA, but without protein deposition, showing that the change was not due to the presence of protein but rather due to the crosslinking of L-DOPA with PAA (Appendix A). The crosslinking of L-DOPA and protein coverage did not affect the stiffness of the stiff gels (which was associated with local changes), as observed in the AFM spectroscopy data. Moreover, the loss factor (G′′/G′) values calculated from LVER were 0.08 and 0.11 for the soft and stiff hydrogels, respectively, demonstrating that the material presents a solid-like character, which is in agreement with the data in the literature on PAA hydrogels [41,42]. These ratios were not affected by the PAA’s functionalization with L-DOPA or collagen deposition (Appendix A).

Investigation of the macroscopic hydrogel surfaces using fluorescently labeled collagen revealed protein aggregates to be occasionally present on the surface (Figure 1E, black). Their occurrence is not dependent on the substrate type. Interestingly, the protein patches (Figure 1E, gray) can be identified as having different sizes in the images. Large, macroscopic patches were observed on glass and stiff hydrogels, while more uniformly spread, small patches were present on the soft hydrogels.

### 3.2. Characterization of LSECs on PAA Hydrogels

After their preparation and characterization, the hydrogels were challenged by seeding LSECs on their surface. The general morphology of the cells seemed to be unaltered when comparing the stiff hydrogel and control (cultured on plastic) substrates (Figure 2), indicating that there was no negative effect of PAA on cell morphology. The cells spread, forming monolayers and groups, depending on local cell density. We did, however, observe a difference in the morphology of LSECs cultured on soft hydrogels. These cells still form large groups, but multiple individual cells of a rounded shape could be observed (Figure 2, soft hydrogel). When AFM measurements were performed on the LSECs forming widespread groups, all cells appeared similar to those cultured on stiff substrates, which were fenestrated and formed tight groups. This observation indicates that soft PAA hydrogels with collagen III substrates are suitable for LSEC culturing, mimicking the soft niche of the LSECs’ natural environment in vivo, with the cells presenting a similar morphology to LSECs cultured on traditional substrates.

In addition to the synthesized hydrogels, we investigated commercial PAA hydrogels with a wide range of elasticities: 0.1, 0.2, 0.5, 1, 2, 4, 8, 16, 25, 50, and 100 kPa. We characterized the changes in the size of the cell area (Figure 3A) and cell nuclei area (Figure 3B) using optical microscopy. The cell area was similar for a wide range of PAA elasticities, exceeding 1400 µm^2^ for hydrogels of >8 kPa, with a gradual decrease below 4 kPa and a sudden drop below 1 kPa (1 kPa–1240 ± 370 µm^2^, 0.5 kPa–682 ± 220 µm^2^, 0.1 kPa–305 ± 60 µm^2^). Moreover, we observed a gradual decrease in the cell nucleus area with the substrate’s softening. Significant changes in cell nucleus size, exceeding 20% (versus glass), were observed for substrates of 4 kPa and softer, which corresponds to the stiffness of a healthy liver [17]. In particular, we observed up to a 45% decrease in the cell nuclei size on the softest substrates compared to that on the glass coverslips.

Interestingly, despite their altered morphology, we demonstrated that the LSECs’ endocytic function was mainly preserved (Figure 3C). For this experiment, we selected three types of hydrogels with elasticities of 2, 8, and 100 kPa. The first two were used to mimic healthy and fibrotic niches, respectively, and the third was a much stiffer PAA control. As a final control, we used standard plastic (polystyrene) multiwell plates. The endocytic capacity of the LSECs (normalized to the number of cells) was similar in all groups and was not dependent on ligand accessibility (Figure 3C). Considering the rounded shape of some of the LSECs cultured on the soft substrate, this result indicates that despite their altered morphology, the LSECs on all substrates were viable and functional. Additionally, we tested 6-keto-PGF1α, a stable metabolite of prostacyclin that serves as a marker of prostacyclin production in endothelial cells and an indicator of endothelial function, inflammation, and vascular homeostasis (Appendix A). We did not observe significant changes in this marker. However, we did observe an insignificant trend towards a lower level of 6-keto-PGF1α on the stiff hydrogels compared to glass coverslips. The levels of 6-keto-PGF1α tend to be higher on soft hydrogels when compared to stiff hydrogels, indicating a mild positive effect of soft substrates on LSEC function.

Recently, we highlighted that fenestration deformability might be an important parameter of LSEC morphology [18]. Here, the deformability parameter was calculated for fenestrations in LSECs cultured on PAA hydrogels. The aim was to use soft PAA to imitate the in vivo space of Disse in the liver sinusoids. This space is localized between LSECs, with their discontinuous basal lamina, and hepatocytes’ microvilli, creating a soft viscous environment. As described in the Section 2 “Materials and Methods”, deformability was calculated as a ratio of the fenestration diameter measured for a high loading force to the diameter calculated for a low loading force during QI AFM imaging and was used to reflect the changes in the pressure and direct forces exerted by flowing blood particles. Our results indicated an increase in the deformability parameter of the fenestrations in LSECs cultured on PAA hydrogels when compared to plastic substrates. The mean deformability parameter (300 pN/170 pN) was equal to 28 ± 20%, 39 ± 25%, and 44 ± 25% for the plastic, stiff hydrogels, and soft hydrogels, respectively. Our results show that using soft hydrogels better mimics the natural environment present in vivo when compared to traditional cell culture substrates.

In order to characterize the influence of soft substrates on the cells’ cytoskeletal architecture, we implemented 3D structured illumination microscopy (3D SR-SIM) (Figure 4). The LSECs cultured on stiff hydrogels were characterized by short and long actin fibers, similar to those in the LSECs cultured on the glass substrate [18,43]. The LSECs cultured on soft hydrogels were characterized by a mesh-like actin structure, which has been reported to be an indicator of healthy, fenestrated LSECs [6,44]. The tubulin network seemed to be unaffected by the cell culture on both hydrogels (Figure 4). Altered tubulin organization is responsible for reduced endocytosis in LSECs [45,46]; therefore, the observed preservation of endocytosis capacity and unaltered tubulin organization are in agreement with previous observations. Interestingly, we observed that the LSECs on soft substrates presented a morphology in which the central part of the cell is lower than the periphery, indicating that the cells were embedded within the hydrogel structure (Figure 4, 3D projection). Cells anchor to the substrate, creating focal adhesion points to the extracellular matrix (ECM) to generate traction forces and extend their cytoskeleton. If the substrate deforms or dissipates stress over time, cells may struggle to establish firm adhesions, reducing their ability to spread effectively. This observation highlights that the viscoelastic properties of the hydrogels were hampering the LSECs’ capability to spread on soft hydrogels.

### 3.3. Characterization of LSECs from a Genetic Model of Systemic Inflammation Cultured on PAA Hydrogels

A recent report by Guixé-Muntet and coworkers indicated that culturing LSECs from a model of liver cirrhosis on a soft substrate (0.5 kPa) resulted in refenestration, i.e., the restoring of fenestrations in defenestrated cells [20]. The authors indicated that nuclear deformation was the signaling mechanism regulating the fenestrations in LSECs. In our report, we demonstrated that LSECs isolated from 6-month-old Mcpip1^fl/fl^ LysM^Cre^ (Mcpip1 KO) mice (a model of systemic inflammation) have a reduced number of fenestrations compared to LSECs from Mcpip1^fl/fl^ (Mcpip1 control) mice and that the defenestration is only partially reversible by cytochalasin B. Therefore, here, we tested whether the culturing of LSECs from this systemic inflammation model effected the refenestration of defenestrated LSECs in vitro when the LSECs were cultured on soft hydrogels. To verify our hypothesis, we cultured the LSECs for 18–20 h on glass and stiff and soft hydrogels to verify the effect on the LSECs’ porosity (Figure 5). Similarly to the observations presented in Figure 1, we observed fenestrated LSECs on all types of substrates in the Mcpip1 control group. Significant defenestration was observed in the LSECs isolated from Mcpip1 KO mice on glass and the stiff hydrogels. The porosity of the LSECs, presented as the number of fenestrations per area, was reduced from 1.55 ± 0.3 fen./µm^2^ (Mcpip1 control) to 0.37 ± 0.17 fen./µm^2^ (Mcpip1 KO) for glass and 0.52 ± 0.24 fen./µm^2^ (Mcpip1 KO) for the stiff hydrogels. Interestingly, no statistically significant differences were found between cells from the Mcpip1 KO and Mcpip1 control mice when cultured on soft hydrogels. Moreover, the number of fenestrations in the LSECs from Mcpip1 control mice were in the same range for all substrates. The average porosity of the cells from Mcpip1 KO mice was 1.13 ± 0.22 fen./µm^2^, while that of Mcpip1 control mice was 1.2 ± 0.35 fen./µm^2^. This shows that fenestrations were restored in the LSECs isolated from Mcpip1 KO mice to the level of control cells. It is worth noting that we additionally observed fenestration-like structures in the nuclear area on soft hydrogels, similar to previous studies [43,47]. These pores, which were the size of fenestrations, were discarded from our analysis. We considered only the pores that gathered in the sieve plates and that could be distinguished as transcellular pores to be fenestrations. The pores in the nuclear area could not perform transcellular trafficking and, therefore, were considered to be fenestrae labyrinths [48].

LSECs isolated from Mcpip1 KO and Mcpip1 control mice were cultured on glass and soft and stiff hydrogel substrates to compare the level of actin polymerization in those cells (Figure 6A,B). We observed more stress fibers in LSECs isolated from Mcpip1 KO mice than in LSECs from the Mcpip1 control group. This change was also seen in LSECs cultured on stiff and soft hydrogels. We detected actin filaments using free FilamentSensor 2.0 [49] software in order to quantify the number of filaments per cell in the LSECs cultured on each substrate (Figure 6B and Appendix A). Similarly to our previous report, we observed an increased number of filaments in the Mcpip1 KO group [18]. The difference in the refractive index of the gel when compared to glass coverslip altered the optical path, resulting in a worsening of the resolution of the fluorescence images. As a result, the overall number of detected filaments in the LSECs cultured on the PAA hydrogels was lower than the number observed for the LSECs cultured on glass substrates. We observed no significant difference between the Mcpip1 control and Mcpip1 KO groups, but the LSECs from Mcpip1 KO mice tended to have less filaments than those of the Mcpip1 controls. We observed an increase in the apparent Young’s modulus values in the knockout groups on all substrates (Figure 6C). The apparent Young’s modulus values for the LSECs in the Mcpip1 control group were equal to 6.1 ± 3.2 kPa, 8.1 ± 3.1 kPa, and 5.8 ± 2.6 kPa, while for the LSECs in the Mcpip1 KO group they were equal and 8.3 ± 5.9 kPa, 9.6 ± 5.1 kPa, and 7.5 ± 4.3 kPa for glass and the stiff and soft hydrogels, respectively. As reported previously, the broader standard deviation in the Mcpip1 KO group might be explained by the subpopulation of LSECs with profound stress fibers [18]. This observation shows that the lower number of actin filaments detected in the LSECs cultured on soft hydrogels in the Mcpip1 KO group did not alter the apparent Young’s modulus of the LSECs in this group.

## 4. Discussion

Mechanotransduction is an important factor regulating the morphology and phenotype of cells. Studies link liver stiffening with an increased risk of cirrhosis with portal hypertension and hepatocellular carcinoma [17,50]. When analyzing the relationship between liver stiffness and the risk of progressing to cirrhosis with portal hypertension, it was observed that for every 1 kPa increase in liver stiffness, the progression rate rose exponentially [50], indicating the importance of studying the influence of soft substrates on liver cell phenotypes. In this study, we demonstrated that LSECs are highly sensitive to substrate stiffness, with soft hydrogels promoting the refenestration of defenestrated cells. By preparing polyacrylamide hydrogels with a thin collagen layer, we aimed to mimic the physiological microenvironment of the liver.

One of the key observations in our work addresses the discrepancy in the apparent Young’s modulus of hydrogels measured using AFM and a rheometer. The Young’s modulus obtained via the rheometer was generally more than 30% lower than the apparent Young’s modulus determined by AFM. This divergence was insignificant for soft hydrogels, but it was significant for stiff hydrogels and observed systematically. These results agree with previous reports highlighting method-dependent variations in hydrogel stiffness measurements [51,52]. AFM measures local, microscale stiffness by indenting the surface perpendicularly with a nano-up-to-microscale probe, typically reaching indentation depths ranging from a hundred nanometers to several micrometers. The calculation of the Young’s modulus assumed Hertzian contact mechanics limited to elastic deformation, which might create errors by neglecting the viscoelastic effects in hydrated hydrogels [41]. AFM alone can lead to discrepancies when using different tip apexes [53]. Indeed, our AFM data from individual maps present high variance (Figure 1C), which is a result of local stiffness differences and the repeatability of the hydrogels’ synthesis. The advantages of such a local approach to the measurements were underscored by Richbourg et al. [54], as the local migration of cells depends on local traction forces. The local variance in hydrogel elasticity may explain the observed clustering of spread cells, while, in other areas, cells remained rounded without spreading or forming their characteristic cobblestone morphology. Our observation is in line with previous observations of LSECs cultured on Matrigel [55], which resulted in the LSECs rounding and forming tubular structures. In contrast, a rheometer applies bulk mechanical stress over a large 8 mm surface area. Rheometric methods provide a shear modulus (*G*′), which is converted to a Young’s modulus, neglecting the viscous contribution from the viscoelastic properties of the material and assuming an ideal isotropic material with a Poisson’s ratio of ~0.5. Hydrogel heterogeneity (in its viscoelasticity and poroelastic properties) could significantly influence bulk measurements, explaining the observed discrepancies. Although AFM and rheometers apply different types of forces (indentation and shear) to PAA hydrogels, the results from these two stiffness measurement methods were generally comparable, with neither method demonstrating clear superiority over the other. The choice of method should depend on the available resources and the specific objectives of the analysis. However, using more than one method to evaluate substrate stiffness provides a better understanding of the viscoelastic effect of the substrate on cell morphology and the effect of gel functionalization on the alteration of those changes [41,52,53]. In particular, as we demonstrated a gradual decrease in nucleus roundness and step-like reduction in LSEC cell area on hydrogels with an elasticity below 4 kPa, we want to highlight that creating and characterizing repeatable hydrogels is an important factor in studying LSECs. Moreover, we showed that gel functionalization with L-DOPA and a protein coating, even when nanometrically thin, alters the stiffness of the hydrogels. The interaction of the two polymers—collagen and PAA—resulted in a non-linear effect on the final cell mechanics that locally increased the stiffness of the soft hydrogels and reduced the stiffness of the stiff hydrogels.

We showed that a soft environment changed the cells’ morphology (reduced cell size and nucleus rounding), but the presence of fenestrations and the LSECs’ function remained unaltered. Control LSECs cultured on all substrates remained fenestrated, and their endocytic properties and production of 6-keto-PGF1α were preserved. This demonstrates the potential of using PAA hydrogels to refenestrate defenestrated LSECs in vitro. These findings remain in agreement with previous studies, which indicated that targeting mechanotransduction pharmacologically (via nesprin) or physically (by culturing LSEC on soft substrates) can benefit LSEC morphology [20,22,56]. Guixé-Muntet et al. concluded that decreasing the stiffness of the LSEC niche is a promising therapeutic target for treating liver fibrosis [20]. In particular, LSEC fenestrations, a hallmark of a healthy liver that enables the efficient clearance of metabolites, lipoproteins, and drugs from the circulation, could be restored by changing the mechanics of the cellular niche. Our results on the porosity of Mcpip1 KO and Mcpip1 control cells cultured on substrates with different specific apparent Young’s moduli provided clear insights about the significant reduction in the number of fenestrations seen in cells from Mcpip1 KO mice when cultured on plastic or stiff hydrogel substrates. This reduction was reversed by culturing LSECs on soft hydrogels. This observation shed a light on the role of mechanotransduction in the progression of liver disease. LSEC fenestrations contribute to immune surveillance by enabling immune cells to extend protrusions through the endothelial layer to detect hepatocellular antigens without extravasation [8,9]. This diapedesis-independent mechanism is crucial for the immune monitoring of infected or transformed hepatocytes. Inflammation-associated sinusoidal defenestration, the reduced deformability of remaining fenestrations, and capillarization impair these interactions, potentially reducing immune regulation. This highlights the broader significance of fenestrations in both metabolic and immunological liver functions. Systemic inflammation in the Mcpip1 KO model results in multiple changes in LSEC gene expression [57], and even though individual markers have been recently highlighted [18], it is not clear which mechanisms lead to LSEC defenestration or refenestration. Our studies showed that the refenestration was not accompanied by significant changes in actin polymerization or a reduced apparent Young’s modulus, similar to other studies, indicating that actin-independent mechanisms regulate LSEC fenestrations [58].

The mechanical properties of PAA hydrogels play pivotal role in LSEC morphology and spreading. On soft substrates, the contact area between the cell and the substrate is reduced due to the LSECs’ interaction with the substrate. The confined effect of their chemical interactions with localized patches of collagen and mechanical interactions are a result of insufficient traction forces, leading to a morphology where the cell center is embedded into the hydrogel, leading to the position of the cell nuclei being lower than the cell’s periphery. The reduced cell area and rounding of the cell nucleus did not affect LSEC porosity and function. Furthermore, Mcpip1 control cells cultured on soft substrates presented a tendency toward reduced actin stress fiber formation, promoting a quiescent and less contractile morphology. A similar tendency has been reported for human umbilical vein endothelial cells (HUVECs), for which soft PAA substrates reduced the contractility of cells and actin labeling intensity in a substrate-stiffness-dependent manner [59]. The authors pointed to reduced Rho/ROCK activity as a potential origin of cell quiescence. Previous studies on LSECs linked fenestration number and size with Rho/ROCK activity [60,61]. The inhibition of ROCK using Y27632 dihydrochloride resulted in an increased number of fenestrations when evaluated using SEM, which aligns with the findings from HUVECs. Except for its effect on fenestrations, ROCK inhibition resulted in the depolymerization of actin filaments in LSECs [59,61]. Our results, however, indicate that Y27632 dihydrochloride did not improve LSEC porosity in Mcpip1 KO mice (Appendix A). Moreover, we did not observe changes in the level of actin polymerization in the Mcpip1 KO group when compared to the Mcpip1 control groups cultured on all substrates. These findings are in line with previous reports showing that other mechanisms, independent of actin, are involved in the regulation of fenestrations in LSECs [18,58]. Nevertheless, further studies are needed to fully understand the complex mechanisms regulating LSEC fenestrations.

## 5. Conclusions

In this study, we demonstrated the synthesis and characterization of PAA hydrogels with defined nanomechanical properties intended to provide an effective biomimetic environment for LSECs, preserving their fenestrated morphology and key functions. We observed that soft hydrogels better mimic the physiological microenvironment of the liver, promoting the refenestration of defenestrated LSECs. Notably, LSECs from a genetic model of systemic inflammation regained fenestrations when cultured on soft hydrogels, underscoring the critical role of substrate stiffness in regulating LSEC morphology and function. As a result, we confirmed that the previously reported refenestration of LSECs observed in a model of liver cirrhosis is also applicable to a genetic model of liver inflammation.

We highlighted that the detailed characterization of hydrogel nanomechanics may vary locally, resulting in distinct cell morphologies. We highlighted the discrepancies between different methods (AFM and rheometry) for measuring hydrogel stiffness and demonstrated how hydrogel functionalization with collagen alters local mechanical properties but has no visible effect on LSECs. Importantly, we found that while the LSECs cultured on soft hydrogels exhibited reduced cell spreading and nucleus rounding, their porosity, endocytosis capacity, and fenestration deformability remained preserved. We concluded that soft PAA hydrogels can be used for culturing LSECs, as they provide a similar morphology and function to standard substrates while positively affecting the actin cytoskeleton, minimizing the presence of actin stress fibers.

Our findings emphasize the potential of soft hydrogels as a promising tool for in vitro LSEC studies and for investigating mechanotransduction-related changes in liver disease. Additionally, the results reinforce our growing understanding that modulating the mechanical properties of the liver microenvironment could serve as a therapeutic strategy for restoring LSEC functionality in liver pathologies such as fibrosis and systemic inflammation.

## Figures and Tables

**Figure 1 cells-14-00621-f001:**
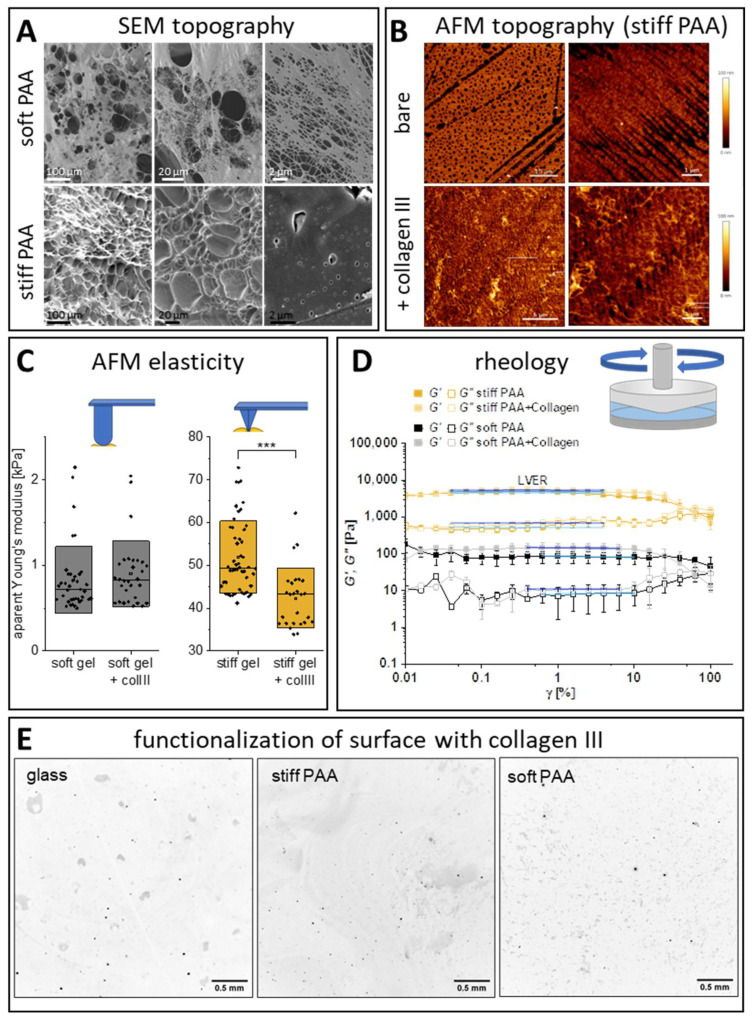
Characteristics of synthesized polyacrylamide (PAA) hydrogels. (**A**) Representative SEM images of dehydrated hydrogels. (**B**) Representative AFM images of the topography of a hydrated stiff hydrogel and stiff hydrogel with collagen III deposited on its surface. Similar images for soft hydrogels are presented in Appendix A. (**C**) AFM spectroscopy showing the distribution of the apparent Young’s modulus for stiff and soft hydrogels, both bare and with 0.1 mg/mL collagen III deposited on their surface. The box range represents the standard deviation, the bar represents the median, and the open square represents the mean value; *** *p* < 0.001. (**D**) Rheological properties of soft and stiff hydrogels. Storage (G′, filled points) and loss (G″, hollow points) moduli were obtained. LVER lines were fit to the plateau of G′ and G″ and used to calculate the loss factor. (**E**) Representative fluorescence images of the collagen III deposited on the surface of glass and stiff and soft hydrogels. Negative images are presented.

**Figure 2 cells-14-00621-f002:**
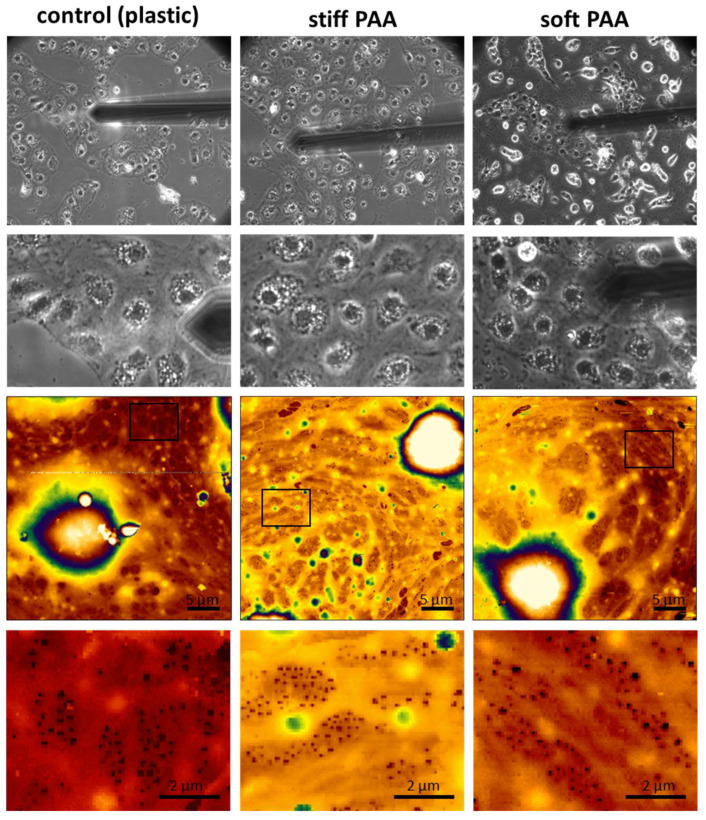
Morphology and ultrastructure of LSECs cultured on plastic and stiff and soft PAA hydrogels. Phase contrast images (top panel) and AFM topography (bottom panel) present a spread of cells with LSEC characteristics, such as the formation of tight groups, a flat cell periphery and bulging nuclei, and sieve plates filled with fenestrations. AFM images are 35 × 40 µm^2^ and 405 × 463 pixels. Insets (corresponding black boxes in each column) represent a digital enlargement of the area with fenestrations. All groups have the same magnification and scale in their corresponding images.

**Figure 3 cells-14-00621-f003:**
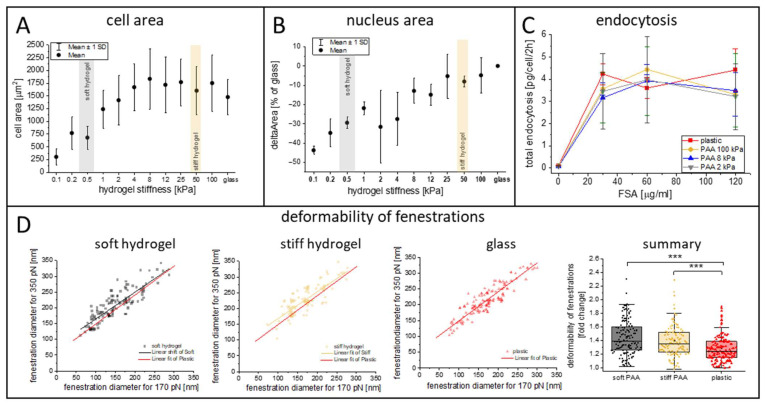
Comparison of changes in cell area (**A**), cell nuclei area (**B**), and endocytosis (**C**) of LSECs cultured on commercial PAA hydrogels with varying elastic properties. The stiffness ranges of the soft and stiff hydrogels are marked in gray and bright orange, respectively. (**D**) The deformability of fenestrations calculated from living LSECs cultured on substrates with varying elastic properties, based on AFM imaging performed according to the methodology described in the Materials and Methods. That summary presents the final deformability parameter, which is calculated as a ratio of the diameter of the fenestrations measured at 300 pN to the diameter measured at 170 pN. More than 100 fenestrations were measured per group in a one-to-one manner; *** *p* < 0.001.

**Figure 4 cells-14-00621-f004:**
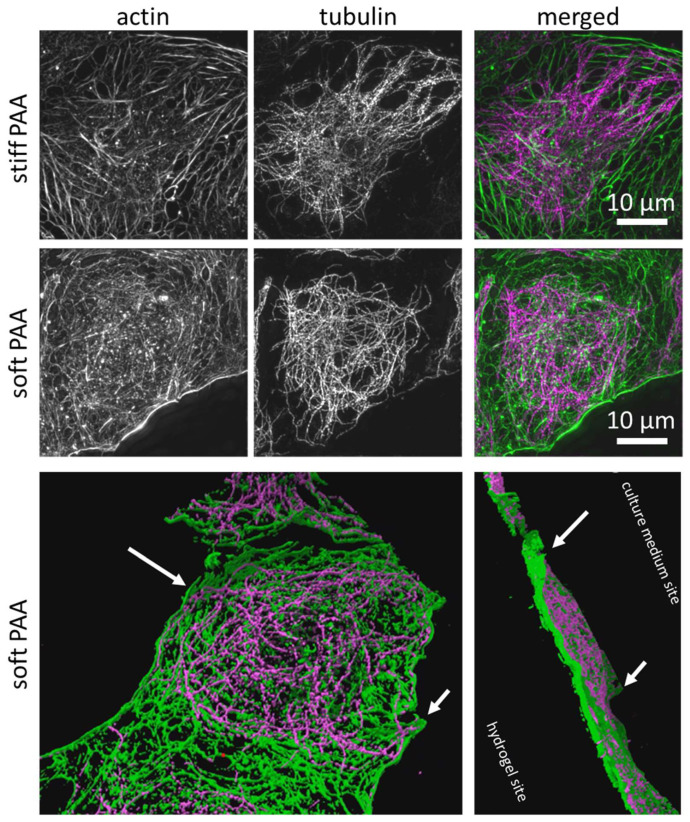
Representative LSECs cultured on stiff and soft PAA hydrogels, visualized using super-resolution structured illumination microscopy (SR-SIM). Actin shows distinct levels of organization, which are translated to an F-actin polymerization degree, and a similar organization of tubulin was seen when LSECs were cultured on stiff and soft substrates. Actin (phalloidin-Alexa Fluor 488, green) and tubulin (immunofluorescence, Alexa Fluor 555, purple) are both visible. The lower panel is a 3D projection (cross-section) of an LSEC cultured on a soft hydrogel, with the elevated edges of the cell, which is embedded within the hydrogel structure (white arrows) (Appendix A). Image size: 40.96 µm × 40.96 µm.

**Figure 5 cells-14-00621-f005:**
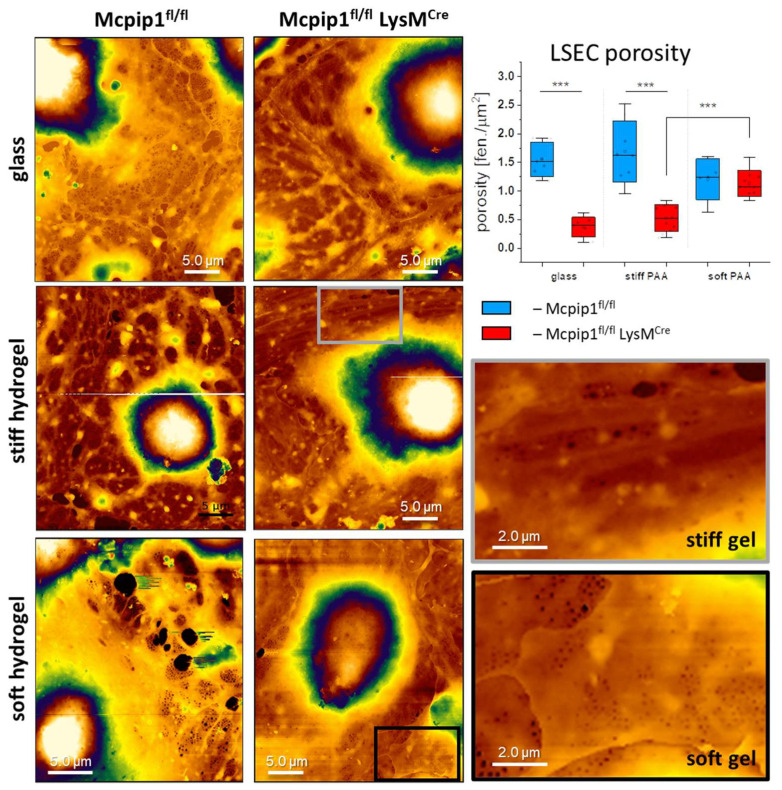
Representative AFM images of LSECs isolated from Mcpip1 control and Mcpip1 KO mice cultured on glass and stiff and soft hydrogels. Selected areas in the cells’ peripheries were digitally magnified to depict fenestrations in corresponding groups (gray square for stiff hydrogel and black for soft hydrogel). Images are presented at the same scale for comparison. Image size: 25 × 30 µm^2^ and 300 × 360 pixels. The chart presents LSEC porosity, expressed as the number of fenestrations per cell area (fen./µm^2^). The box range represents the standard deviation, the bar represents the median, and the open square represents the mean value; *** *p* < 0.001.

**Figure 6 cells-14-00621-f006:**
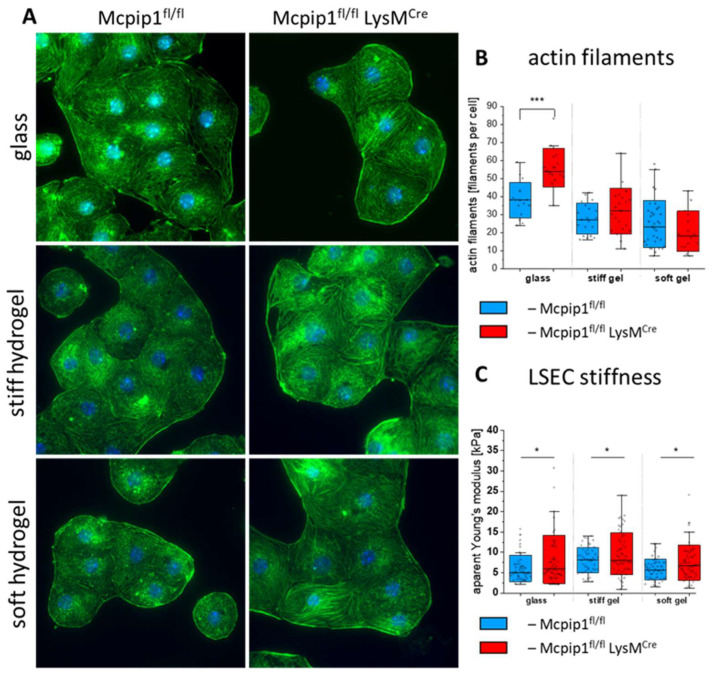
(**A**) Representative fluorescence images of actin and cell nuclei of LSECs isolated from Mcpip1^fl/fl^ and Mcpip1^fl/fl^ LysM^Cre^ mice and cultured on glass and stiff and soft PAA hydrogels. Image size: 133.1 × 133.1 µm^2^. (**B**) Actin filaments detected using free FilamentSensor 2.0 software. Representative images containing detected filaments are presented in Appendix A. (**C**) Cell stiffness of LSECs expressed as apparent Young’s modulus and calculated for each group. Box range represents standard deviation, the bar represents the median, and the open square represents the mean value; * *p* < 0.05, *** *p* < 0.001.

**Table 1 cells-14-00621-t001:** Composition of polyacrylamide hydrogels.

	Acrylamide Concentration [%]	Bisacrylamide Concentration [%]	Ingredient Quantity for 1 mL of Hydrogel [µL]		
			Acrylamide (40%)	Bisacrylamide (2%)	Deionized Water
Stiff hydrogel	12	0.25	300	125	575
Soft hydrogel	3	0.6	75	300	625

## Data Availability

The raw data supporting the conclusions of this article will be made available by the authors on request.

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
