# Peer review of "Mimicking the Liver Sinusoidal Endothelial Cell Niche In Vitro to Enhance Fenestration in a Genetic Model of Systemic Inflammation"

_cells, 2025, doi:10.3390/cells14080621_

Round 1

Reviewer 1 Report

Comments and Suggestions for Authors
  1. AFM and rheometer-derived Young’s moduli differ significantly (e.g., stiff hydrogels: 52 kPa vs. 26.3 kPa). The lack of a unified explanation (e.g., scale dependency, viscoelasticity) undermines confidence in substrate reproducibility.

Figure 1A: SEM images of dehydrated hydrogels may misrepresent hydrated microstructures, as dehydration collapses pores. Hydrated AFM imaging (Fig. 1B) for soft gels is described as "hampered," raising concerns about accuracy. 

  1. To enhance the background on wound healing, integrating references such as [10.1002/advs.202405463] (detailing advanced biomaterial strategies) and [10.1016/j.colsurfb.2014.10.057] (exploring interfacial dynamics in tissue repair) would contextualize the study’s relevance.

  1. Figures 2, 5: Fenestration diameters are manually measured via AFM, introducing bias. Automated algorithms (e.g., machine learning-based segmentation) would improve objectivity.

Insets in AFM images lack scale bars, complicating validation of fenestration size (50–350 nm range). 

  1. : Actin filament counts using FilamentSensor 2.0 are affected by hydrogel opacity, yet resolution limitations are not quantitatively addressed. This may skew conclusions about actin’s role in fenestration regulation.

  1. The endocytosis assay (Fig. 3C) shows preserved function but lacks temporal resolution. Time-course data linking fenestration recovery to functional restoration would strengthen claims.

  1. Figure 4 (3D SIM): LSECs appear embedded in soft gels, but the study does not disentangle mechanical embedding from biochemical signaling (e.g., collagen III distribution in Fig. 1E). Heterogeneous protein patches could confound stiffness-specific effects.

Author Response

Please see the attachment. As some of the remarks from other Reviewers overlap, we addressed all of the remarks in one document. We identify you as Reviewer 1. 

Reviewer 2 Report

Comments and Suggestions for Authors

The article “Mimicking the Liver Sinusoidal Endothelial Cell Niche in vitro 2 to Enhance Fenestration in a Genetic Model of Systemic In-flammation .

Although the study is valuable, it has some shortcomings.

Abstract: Modify the abstract and write methods in abstract

Introduction

Shorten and Modify the introduction by adding new references.

Last paragraph should be rewritten

Methods

Methods should be mentioned with references

Results:

Rewrite all the results as they are clear

Conclusion should be written with clear understanding to readers

MINOR COMMENTS

Various situations should be considered that will increase the research value.

Add limitation of the study

Grammatical Errors should be removed. Typos should be corrected.

The article should be accepted after MAJOR revision.

Comments on the Quality of English Language

Grammatical Errors should be removed. Typos should be corrected.

Author Response

Please see the attachment. As some of the remarks from other Reviewers overlap, we addressed all of the remarks in one document. We identify you as Reviewer 2. 

Reviewer 3 Report

Comments and Suggestions for Authors

This manuscript presents a well-executed and timely study investigating how mechanical cues influence the morphology and cytoskeletal dynamics of liver sinusoidal endothelial cells (LSECs) under inflammatory conditions. By using polyacrylamide hydrogels with tunable stiffness and primary LSECs, the authors provide valuable insights into the interplay between systemic inflammation, endothelial mechanosensitivity, and fenestration architecture. The findings are highly relevant to researchers studying liver microenvironment remodeling, endothelial mechanobiology, and inflammatory liver disease. However, a few critical aspects require clarification or further investigation before the manuscript can be recommended for publication:

  1. While the study focuses on LSECs, the isolation protocol and purity assessment (e.g., Fig. 1A–B) need to be described more clearly. It is not evident whether other liver endothelial subtypes or contaminating cell populations were effectively excluded from the cultures. Clarifying the gating strategy and the specificity of the markers used would significantly enhance the rigor and reproducibility of the findings.
  2. The authors show actin cytoskeletal changes across substrates of varying stiffness (Fig. 2), but the quantification is only briefly addressed. A more detailed analysis of F-actin intensity or stress fiber organization would strengthen the conclusions regarding cytoskeletal tension and mechanical adaptation, especially in the context of inflammatory stress.
  3. Physiological Relevance and Context of Fenestration Loss. The manuscript briefly mentions the role of LSEC stiffness in fibrosis progression, but the discussion could be expanded to better contextualize the in vitro findings within the in vivo liver microenvironment. In particular, it is disappointing that the introduction and discussion reduce the role of LSEC fenestration to general metabolic trafficking, without acknowledging recent breakthroughs in its relevance to immune responses (PMID: 26188075, 25892224) during fibrosis, HBV infection and progression of metabolic dysfunction-associated steatohepatitis (MESH). This point is critical for the translational relevance of the study and better position the work within current literature.

Author Response

Please see the attachment. As some of the remarks from other Reviewers overlap, we addressed all of the remarks in one document. We identify you as Reviewer 3. 
